# Collaborative Alignment of NLP models

**Fereshte Khani**
Microsoft
fkhani@microsoft.com

**Marco Tulio Ribeiro**
Google DeepMind*
marcotcr@gmail.com

## Abstract

Despite substantial advancements, Natural Language Processing (NLP) models often require post-training adjustments to enforce business rules, rectify undesired behavior, and align with user values. These adjustments involve operationalizing "concepts"—dictating desired model responses to certain inputs. However, it's difficult for a single entity to enumerate and define all possible concepts, indicating a need for a multi-user, collaborative model alignment framework. Moreover, the exhaustive delineation of a concept is challenging, and an improper approach can create shortcuts or interfere with original data or other concepts.

To address these challenges, we introduce CoAlign, a framework that enables multi-user interaction with the model, thereby mitigating individual limitations. CoAlign aids users in operationalizing their concepts using Large Language Models, and relying on the principle that NLP models exhibit simpler behaviors in local regions. Our main insight is learning a *local* model for each concept, and a *global* model to integrate the original data with all concepts. We then steer a large language model to generate instances within concept boundaries where local and global disagree. Our experiments show CoAlign is effective at helping multiple users operationalize concepts and avoid interference for a variety of scenarios, tasks, and models.

## 1  Introduction

NLP models have showcased remarkable capabilities, yet they are not exempt from imperfections. Unacceptable values embedded in their training data, persistent errors, or violations of business rules highlight the need to teach certain *concepts* to these models. A concept relates a set of inputs to desired behaviors, e.g. in sentiment analysis, a concept may dictate that "religion does not connote sentiment" (e.g., "I'm Muslim" is Neutral). Similarly, in natural language inference (NLI), the broader concept of "downward monotonicity" specifies entailment relations when certain parts of expressions are made more specific (e.g., "All cats like tuna" entails "All small cats like tuna").

The standard way of teaching concepts to models is adding new training data that exemplifies the concept, e.g. adding neutral sentences containing religious words [1], or adding entailment pairs that illustrate downward monotonicity [2]. However, it is hard to ensure that the data provided does not lead to *shortcuts*, i.e. spurious correlations or heuristics that allow models to make predictions without capturing the true underlying concept, such as "all sentences with religious terms are neutral", or "going from general to specific leads to entailment". Further, the model may *overfit* – fail to generalize from the instances provided to the actual concept, e.g. only recognizing pairs in the form ("all X...", "all ADJECTIVE X..."), and not pairs like ("all animals...", "all cats..."). Finally, both shortcuts and overfitting can lead to *interference* with the original data or other concepts, e.g. leading to failures on "I love Islam" or pairs like ("Some cats like tuna", "Some small cats like tuna"). In sum, operationalizing concepts is challenging, since users typically cannot think of all concept boundaries and interactions in advance.

---

*Work done while at Microsoft

37th Conference on Neural Information Processing Systems (NeurIPS 2023).

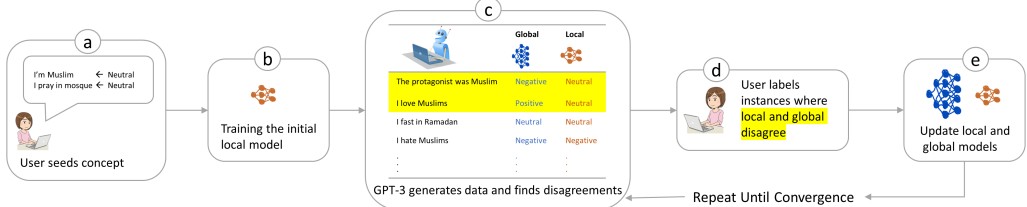

Figure 1: CoAlign loop for operationalizing a single concept. (a) The user starts by providing some seed data from the concept and their labels, (b) they are used to learn a local concept model. (c) GPT-3 is then prompted to generate new examples, prioritizing examples where the local model disagrees with the global model. (d) Actual disagreements are shown to the user for labeling, and (e) each label improves either the local or the global model. The loop c-d-e is repeated until convergence, i.e. until the user has operationalized the concept and the global model has learned it.

One possible solution is to ask domain experts to create data that covers the concept as comprehensively and precisely as possible, e.g. the GLUE diagnostics dataset [3] or the FraCaS test suite [4]. However, these datasets are often expensive to create, small (and thus not suitable for training), and not exhaustive, as even experts still fail to think about all aspects and nuances of a concept [5]. Another solution is to have users provide data incrementally while they receive feedback from the model, e.g. adversarial training [6] or adaptive testing [5]. These do not require users to think about everything in advance, and can expose and correct model weaknesses. However, adversarial training does not include the notion of concepts explicitly, and adaptive testing does not address the interaction between different concepts or between a concept and the original data. As a result, users may not explore concept *boundaries* efficiently, and thus do not know when they have sufficiently covered a concept or whether they have introduced interference with other concepts.

In this paper, we introduce Collaborative Alignment of NLP Models (CoAlign). CoAlign leverages the collective knowledge of multiple users to cover many concepts instead of relying on a single user. CoAlign allows users to collaborate with AI systems *and* other users in specifying concepts and teaching them to models. Our key insight is making concepts explicit by learning a *local* model for each concept, in addition to a *global* model that integrates the original data and all additional concepts. When operationalizing a new concept, we rely on a large language model (LLM) to generate new instances where the local and global model disagree, and ask users to label such examples (Figure **??**). Intuitively, these examples are either cases where the local model is not yet fully specified, or where the global model still makes errors on the concept (due to overfitting or shortcut reliance). As users label these examples, both models are updated until convergence, i.e. until the concept has been learned in a way that does not conflict with prior data or prior concepts. In a sense, each local model is ever-improving cheap expert in its respective concept. The speed of prediction of local models and diversity of examples generated by the LLM enable users to explore the boundaries between concepts and existing data, an exploration that could be challenging for users to do without aid.

Our experimental results demonstrate the effectiveness of CoAlign in operationalizing concepts and handling interference. We first show CoAlign outperforms AdaTest [5], a SOTA tool for debugging NLP models that also uses GPT-3 by revealing more bugs and fixing them without interference. We then demonstrate that CoAlign operatinalize concepts even when user starts with biased data, outperforming a model that relies solely on data collection. We compare the data selection mechanism of CoAlign to random selection and uncertainty sampling by running simplified version of CoAlign where instead of using GPT-3 we iteratively select examples from an unlabeled pool. We show CoAlign outperforms both baselines in teaching an NLI model about downward- and upward-monotone concepts [3], as well as teaching a sentiment analysis model about Amazon products reviews. Finally, in a pilot study, we demonstrated that CoAlign helped users clarify their concepts.

## 2 Setup

Let $x$ be a text string, and $y$ a categorical label, e.g. sentiment (positive, negative, neutral). We assume there is a true function $f(x) = y$ that we aim to approximate with a model $\hat{f}(x)$. Assume

we have access to a "base" dataset $D_0 = \{(x_1, y_1), \ldots, (x_n, y_n)\}$, e.g. of movie reviews, from base distribution $P_0$. We refer to the model trained on $D_0$ as the base model $\hat{f}_0$.

A concept $C_i$ is associated with a distribution $P_i$ over the input space, e.g. a concept might place probability mass exclusively on sentences containing religious words. We say $x \in C_i$ if $P_i(x) > 0$. Since it's hard for users to be exhaustive, we do not assume users can generate from $P_i$, but that they can label any $x$ with $f(x)$, and as to whether it is in $C_i$ or not. We further assume users can provide a very small number samples in the support of $P_i$.

Without loss of generality, we assume that we have $k$ users developing a model collaboratively, each with their own concept. Our goal is to train a model $\hat{f}$ that does well on both the base distribution and all concepts, i.e., minimizing $\frac{1}{k+1} \sum_{i=0}^{k} \mathbb{E}_{x \sim P_i}[\hat{f}(x) \neq f(x)]$.

## 3 Collaborative Alignment of NLP models

In this section we describe how a single user operationalizes their concept by producing a dataset $D_i$ in the context of an existing global model $\hat{f}$ trained on the base dataset $D_0$ and previous concepts $D_{1:i-1}$. If $\hat{f}_0$ is already "aligned" within the concept, i.e. $\hat{f}_0(x) = f(x)$ for all $x \in C_i$, we would be done and there would be no need for $D_i$. Thus, what we really want is for $D_i$ to specify the boundary around "failures", cases not currently handled by $\hat{f}_0$. We abstract away the choice model and learning procedure, assuming that the model can learn the concept given $D_i$ (in practice, we use neural networks that are expressive enough).

### 3.1 Sampling from the concept

Since by assumption we cannot sample from the concept distribution $P_i$, it is a challenge to find regions of $P_i$ where $\hat{f}_0$ fails. To address this, we use GPT-3 [7] as a generator $\mathcal{G}$ to simulate a random walk within the concept. To do so, we construct a prompt with $m$ in-concept examples, and use this prompt as input to $\mathcal{G}$ to generate more samples. Then, we ask the user in the loop to accept or reject each generated sample $x'$ ($x'$ is accepted if $x' \in C_i$), and also to label the accepted $x'$ with $f(x')$. The value of $m$ controls the tradeoff between precision and recall, with high $m$ generating more in-concept examples and low $m$ exploring the space more broadly.

Under some conditions it can be shown that $\mathcal{G}$ simulates a Markov chain with stationary distribution $P_i$ (Appendix A), but the weaker condition of connectivity suffices for finding the concept failures, i.e. there must be a path between any $x', x'' \in C_i$ with nonzero transition probabilities according to $\mathcal{G}$ and the prompt. That is, if the concept is connected, with enough time we should be able to find regions of $P_i$ that are not already learned by $\hat{f}_0$.

While sampling from $\mathcal{G}$ *eventually* leads to the yet-unlearned concept regions, it is an inefficient use of human effort, as it does not use the user labels to guide generation (i.e. the user has to label many examples that the current model already handles correctly). A better approach would be ask the user to label in a way that maximizes the expected information gain for the concept, to which we now turn.

### 3.2 Local Concept Models

Complex functions can be approximated by simpler functions in a local neighborhood, as evidenced by theoretical results (e.g. Taylor expansion) and empirical applications [8, 9]. Since a concept is a natural local neighborhood, we use this insight and learn a *local* model $\hat{f}_i$ to approximate $f(x)$ in $C_i$.

We present a toy example for intuition in Figure 2, where we show toy distributions $P_0$ (Figure 2a) and $P_0$ with an additional concept $P_1$ (Figure 2b). In 2b, $\hat{f}_0$ learned on samples from $P_0$ (dashed line) predicts most of $P_1$ correctly, except for a small region in the bottom left quadrant. However, we would need *many* random samples from $P_1$ in order to learn that region, and reach the best model $\hat{f}$ (solid line in Figure 2b). In contrast, we can learn a good local model $\hat{f}_1$ for $P_1$ with a trivial number of samples (Figure 2c). This local model can be used to produce a *disagreement region* between $\hat{f}_1$ and $\hat{f}_0$ (Figure 2d), and sampling from that region would lead to failure discovery much faster.

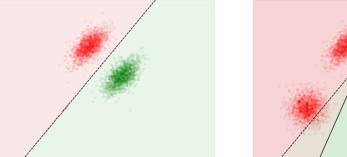 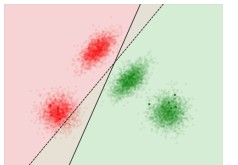 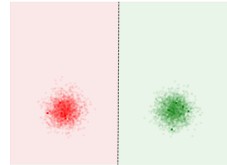 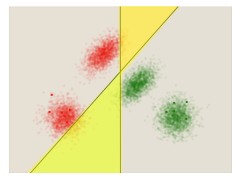

(a) The initial data ($P_0$) is elliptical Gaussian  (b) The user concept ($P_i$) is spherical Gaussian  (c) local function for the user data  (d) disagreement regions between local and global models

Figure 2: (a) A model is trained on two elliptical Gaussian (b) The spherical Gaussian are showing a user concept, who wants to find bugs in the model and teach the model about the new concept. However, since the model (dashed line) have high accuracy on the user concept it is hard to find bugs and teaching the model requires data from the low probability region. (c) We fit a classifier to the Spherical Gaussian which can be done with only a few data points (i.e., local model), (d) we then focus on disagreements between these two models to find bugs in the user concept.

More generally, we define a score function as the disagreement between the local and global function. This score function is used to steer generation such as to maximize the score of generated samples $x'$, by adding instances to the prompt for $\mathcal{G}$ with probability proportional to their score (similar to Ribeiro and Lundberg [5], who use a different score function). We note that models may present false agreement on some samples, i.e. $\hat{f}_i(x') = \hat{f}_0(x') \neq f(x')$. To account for this, we also sample from the agreement region sometimes, with a probability that decays over time as we gain more confidence in the local model.

### 3.3 Operationalizing a concept: from disagreement to convergence

The local and global models disagree on regions where the concept has not yet been learned (global is wrong) or the local model does not fit the user's concept correctly (local is wrong). Thus, every label from the disagreement region results in an improvement in whichever model is incorrect. As labeling progresses, we update both the concept model $\hat{f}_i$ and the global model $\hat{f}$ until we reach convergence (i.e. $\hat{f}$ has learned the concept $C_i$). Note that while $\hat{f}_i$ is trained on data from $P_i$, $\hat{f}$ is trained to do well on the base distribution $P_0$, as well as on all concepts $P_{1:k}$.

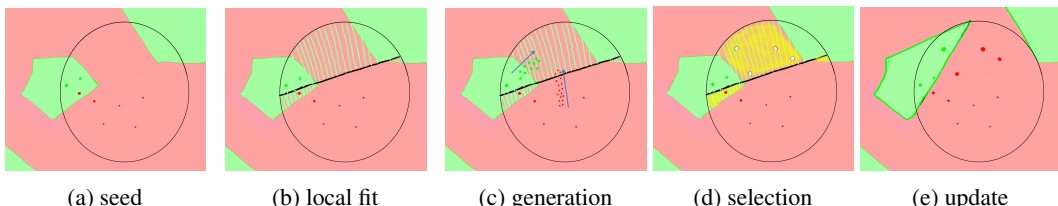

(a) seed          (b) local fit          (c) generation          (d) selection          (e) update

Figure 3: This figure illustrates the main steps in CoAlign: (a) The user starts by providing a small number of data points and their labels within their domain (represented by the black circle), (b) we fit a simple model (shown as a linear model) to represent the user's concept, (c) we use the generator ($\mathcal{G}$) to generate data points towards the region where the local model disagrees with the global model, (d) a diverse set of data points are selected for the user to label, (e) based on the user's feedback, the local and global models are updated until convergence.

We present pseudo-code for operationalizing a concept in Algorithm 1, and illustrate it in Figure 3. The local and global models are initialized in lines 1-2 (Figure 3a and b). In line 5 (Figure 3c), $\mathcal{G}$ is prompted such as to generate in-concept samples that are likely to be in the disagreement region. Then, in line 6 (Figure 3d), we select a batch of $b$ examples for labeling according to the disagreement score of the generated instances. This generation-labeling loop is repeated for $L$ iterations, after which both the local and global models get updated in lines 7-8 (Figure 3e). For the sake of interactivity, in practice we do not train from scratch when updating the models, and instead finetune them from a previous checkpoint. After a few iterations, $\hat{f}$ and $\hat{f}_i$ converge (i.e. the user has operationalized the concept and the global model has learned it). If the number of generated samples $b * L$ is large

enough, we can assume that no disagreement on generated sample between $\hat{f}$ and $\hat{f}_i$ means they have converged, and thus we stop and output the generated data $D_i$ (line 9).

---

**Algorithm 1:** Operationalizing a new concept $i$

**input** : Base dataset $D_0$, Concept datasets $D_{1:i-1}$, A small set of samples from concept $D_i$

1 **Init local and global:** Train $\hat{f}_i$ on $D_i$, and train $\hat{f}$ on $D_{0:i-1}$;

2 **do**

3     **for** $L$ *iterations* **do**

4         **Generation:** Prompt $\mathcal{G}$ with subset from $D_i$ chosen with probability $\propto |\hat{f}(x) - \hat{f}_i(x)|$ ;

5         **Labeling:** Select $b$ samples with prob. $\propto |\hat{f}(x') - \hat{f}_i(x')|$. Users reject $x'$ if out of concept, or add to $D_i$ with label $\hat{f}(x')$;

6     **Update local and global:** Train $\hat{f}_i$ on $D_i$, and train $\hat{f}$ on $D_{0:i}$;

7 **while** $D_i$ *was updated this round*;

**output :** A dataset $D_i$

---

## 3.4   Handling interference between concepts

When aligning a model to multiple users (or training a model on different distributions), two types of conflicts can happen: (1) Literal disagreements: where two users disagree in labeling the same input. (2) Interference: where over-generalization in one concept interfere with another concept (e.g., if a user adds a lot of neutral examples for the "religion does not connote sentiment" concept, then training the model on these new examples can cause the model to predict neutral for sentences like "I love Islam" which interfere with a concept that says "love X" is positive). As stated in Section 2, we assume there is a single true function $f$ (thus no literal disagreement) and we only focus on handling interference. [2] Handling interference is crucial since any local change in ML models can interfere with other parts of the model [10–12].

Having local functions (cheap experts) enable us to check interference efficiently. Every time that a user operationalizes a concept according to Algorithm 1, we check the resulting global model $\hat{f}$ against the local models for all previous concepts. To do so, we re-run Algorithm 1, only asking the user for labels if there is a newfound disagreement region. In practice, this means that a user adding a new concept needs to make sure it does not break any concepts from other users, a process similar to regression testing in software engineering.

While we want to handle interference, having multiplicity of concepts can be beneficial in refining a concept. In particular, we want the global model to not overfit to a concept (i.e., only memorize the training data) and generalize well; however generalizing well is dependent on other concepts and the previous data. Take, for instance, a world where only bananas are yellow. A user might guide an ML model to recognize bananas solely based on their yellow color. Now if a new user introduces another yellow object, like corn, the model must discern other distinguishing features. Merely combining training data for two such concepts doesn't suffice (as shown in [11]); the boundaries must be distinguished. Furthermore, interference can be beneficial by exposing false agreement regions between the global model and any individual concept model. In other words, while both $\hat{f}$ and $\hat{f}_i$ may be relying on the same shortcut to solve some regions of $P_i$, it is unlikely that this shortcut does not cause interference between $\hat{f}$ and all local models $\hat{f}_j$ for $j \neq i$. In this case, the interference caused by adding $C_j$ is actually beneficial, as it further refines concept $C_i$.

In practice the original dataset ($D_0$) is often very large and we cannot fine-tune the model on $D_{0:k}$. Instead of choosing the whole $D_0$, every time we sample data points with highest disagreement between $\hat{f}_0$ and $\hat{f}$ from $D_0$. In other word, we treat $\hat{f}_0$ as a user with concept distribution $P_0$, this enable us to deal with interference with original model as well (as an example in Figure 2 we might only choose the data points from elliptical Gaussian in disagreement region).

---

[2] We like to note that while we haven't introduced a specific mechanism to resolve literal disagreements, our method can surface such disagreements. This can pave the way for resolution through discussions, voting, or even tweaking the model to reflect multiple perspectives, especially in cases where there isn't a consensus among users.

# 4 Linear regression analysis

We now study CoAlign in a noiseless overparametrized linear regression scenario. This particular setup is used in recent literature to gain some insights into the behaviors of deep networks [10, 11, 13, 14]. For more details and proofs see Appendix B.

**Setup.** Let $x \in \mathbb{R}^d$ where only a subset the data points are valid. We assume $\theta^\star \in \mathbb{R}^d$ such that $y = {\theta^\star}^\top x$. Let $S_i$ denote the smallest subspace containing all valid data points of $C_i$. Given $k$ examples in $C_i$, let $S_i^{obv}$ denote the subspace observed by the training data, $S_i^{uno}$ denote the unobserved subspace (thus $S_i = S_i^{obv} + S_i^{uno}$), and $S_i^{inv}$ is the subspace that concept $i$ does not have any variation in it. We consider overparametrized noiseless linear regression, where the number of features ($d$) exceeds the number of training data, enabling us to always interpolate all training data. We assume local and global models infer the min $L_2$ norm interpolant.

As a running example, let $x \in \mathbb{R}^3$ and consider a concept where data points belonging to that concept satisfies $x_1 = x_2$. Let's assume we observed $x = [1, 1, 0]$ with label $y = 2$. In this case we have: $S_1^{obv} = \{[1, 1, 0]\}$, $S_1^{uno} = \{[0, 0, 1]\}$ and $S_1^{inv} = \{[1, -1, 0]\}$. The min-norm solution interpolating the concept is $\hat{\theta}_i = [1, 1, 0]$.

**Operationalizing a concept.** The min-norm solution can also be constructed by introducing constraints that projection of $\hat{\theta}_i$ on $S_i^{uno}$ and $S_i^{inv}$ has to be zero. In the above example, the min-norm solution is the unique answer of the following linear equations ($[0, 0, 1]\theta = 0, [1, -1, 0]\theta = 0, [1, 1, 0]\theta = 2$). When we teach the global model about a concept, the naive combination of data points could violate these constraints, leading to disagreements between local and global models. For our running example, suppose the original data point is $x = [0, 1, 1], y = 2$. Combining the concept data point with the original data point for inferring global model results in $\hat{\theta}_{\text{global}} = [\frac{2}{3}, \frac{4}{3}, \frac{2}{3}]$, which leads to a disagreement between local and global model for data points varying in the $[0, 0, 1]$ direction.

In case of disagreement, two scenarios could arise: (1) the direction is non-zero and the local model needs more specification (happens a lot at the beginning), or (2) the direction is zero, but an explicit constraint is needed to prevent the global model from assuming other values. In our running example, the generator can help us to find a data point that the two model disagree [3]. Let's assume the data point is $x = [0, 0, 1]$ where local model predicts $0$ but global model predicts $\frac{2}{3}$. We show this data point to the user and either the user confirms $[0, 0, 1]$ as non-zero or makes the zero explicit, resulting in a global prediction of $\hat{\theta}_{\text{global}} = [0, 2, 0]$.

Once we learn the local concept (i.e., all unobserved directions are indeed zero), how many of $S_i^{uno}$ directions need to be added as explicit constraints? Intuitively, we do not need to query user for any direction in $S_i^{uno}$ that has already been observed in $S_0^{obv}$ or are orthogonal $S_0^{obv}$ since there is no interference. The following proposition indicates the maximum number of disagreements for teaching global model about the local concept.

**Proposition 1.** *If* $\text{proj}_{S_i^{uno}}(\theta^\star) = 0$, *then the maximum number of disagreement between local and global models is* $\dim(\text{proj}_{S_0^{obv}}(S_i^{uno} \cap (S_i^{uno} \cap S_0^{obv})^\perp))$.

**Handling interference between concepts.** During the operationalization of concept $i$, we maintain $S_0^{obv}$ unchanged. However, in handling interference (Section 3.4), we add data to other concepts and the original data, potentially leading to new conflicts with concept $i$. Therefore, in addition to considering the projection of $S_i^{uno}$ on observed subspaces we need to consider the unobserved subspaces as well. Similar to above, if an unobserved direction in one concept has been observed in another concept we do not need to query user. With notation of $S_{0:k}^{obv}$ denoting sum of all the $S_i^{obv}$, and $S_{-i}$ denotes sum of all subspaces except $i$, the following proposition bounds number of times users need to add data to their concepts due to interference.

**Proposition 2.** *If for all* $i$, $\text{proj}_{S_i^{uno}}(\theta^\star) = 0$ *then the maximum number of times that we need to handle interference is* $\sum_{i=1}^{k} \dim\left(\text{proj}_{S_{-i}}\left(S_i^{uno} \cap (S_i^{uno} \cap S_{0:k}^{obv})^\perp\right)\right)$.

---

[3] Recall that only some of the data points are valid thus we need the generator to find a data points with variation in $[0, 0, 1]$ direction.

| Concept | Examples | Examples of bugs found by CoAlign | | |
|---|---|---|---|---|
| X person = not antonym (X) person | How can I become a positive person? How can I become a person who is not negative? | predicts duplicate shortcut bugs | How can I become a mysterious person? How can I become someone with no mystery? | |
| | | predicts non-duplicate overfit bugs | How can I become a blind person? How can I become someone who has lost his (physical) vision? | |
| Modifiers changes question intent | Is Mark Wright a photographer? Is Mark Wright an accredited photographer? | predicts not-duplicate shortcut bugs | Is he an artist? Is he an artist among other people? | |
| | | predicts duplicate overfit bugs | Is Joe Bennett a famous court case? Is Joe Bennett a famous American court case? | |

Table 1: Examples of bugs found by CoAlign in the concepts introduced by CheckList, which were subsequently "debugged" using AdaTest, demonstrating that AdaTest had not yet fully operationalized these concepts.

| | $C_{orig}$: "X = not antonym (X)", $C_{new}$: "Modifiers changes question intent" | | $C_{orig}$: "X = synonym (X)", $C_{new}$: "less X = more antonym (X)" | |
|---|---|---|---|---|
| | CoAlign | AdaTest | CoAlign | AdaTest |
| broken by new concept | 7/50 | 24/50 | 9/50 | 18/50 |
| fixed by new concept | 5/50 | 2/50 | 20/50 | 18/50 |

Table 2: Comparison of CoAlign and AdaTest in terms of handling interference. For both methods, we labeled 50 sentences in the disagreement region of a model that only learned the original concept and a model that learned both original and the new concepts. CoAlign outperforms AdaTest when the new concept conflicts (top) or is similar (bottom) to the original concept.

## 5  Experiments

We first show CoAlign outperforms AdaTest [5], a leading NLP debugging tool that also uses GPT-3, by revealing more bugs and resolving them without interference. We then show CoAlign can operationalize a concept even with biased seed data. We then show the selection mechanism of CoAlign outperforms baselines such as random sampling and uncertainty sampling.We conclude with a small pilot study of using CoAlign.[4]

**CoAlign outperforms AdaTest[5].**   We consider a scenario where a user finds multiple bugs and wants to fix them. Following Ribeiro and Lundberg [5], we use Quora Question Pairs (QQP) dataset where the goal is to predict if two questions are duplicate or not. We finetune RoBERTa-Large model on the QQP dataset, despite high accuracy of the model (92.2%), Ribeiro et al. [15] identified multiple concepts where the model has low performance. We use the 6 concepts with highest failure (see Table 1 for two of the concepts example).

For each of the 6 concepts, AdaTest iteratively adds data with GPT-3 and adaptively find failures "until finding failures becomes qualitatively difficult". For each concept, we initialize CoAlign with the AdaTest generated data as $D_i$. Even though the model has been 'debugged' with AdaTest, CoAlign quickly reveals $\approx 5$ semantically meaningful sub-categories of bugs for each concept (with many failures within each sub-category). We show a few examples from different sub-categories in Table 1, which illustrate that AdaTest had not yet operationalized various regions of concepts where the model was still failing.

We now compare CoAlign with AdaTest in terms of handling interference. To do so, we pick pairs of concepts $C_{orig}$, $C_{new}$ that might cause interference with one another, but that were debugged and reported as "fixed" by Ribeiro and Lundberg [5]. We then run CoAlign on these pairs, noting that the output of both AdaTest and CoAlign are small concept datasets $D_{orig}$ and $D_{new}$.

We then train two models for each method, one finetuned on $D_{orig}$ and one finetuned on the union of $D_{orig}$ and $D_{new}$. Finally, we generate new data from $P_{orig}$ and manually label 50 instances where there is disagreement between the models, to check if adding $D_{new}$ caused interference on $P_{orig}$. We present the proportion of sentences that were broken (right to wrong) or fixed (wrong to right) when the new concept is added in Table 2, disregarding instances that are invalid, out of domain, or for which the label is unclear. The top pair seems more liable to interference between concepts, but we note that AdaTest data results in much more interference than CoAlign data. In the bottom pair,

---

[4]Code and data will be released in `https://github.com/fereshte-khani/CoAlign`.

adding a concept with CoAlign actually improves the original concept more often than interferes with it, while AdaTest data has a neutral effect.

|  | biased-SB | SB |
|---|---|---|
| Base | 86.7 ± 2.5 | 82.6 ± 1.7 |
| Data collection | 98.6 ± 0.9 | 80.7 ± 1.6 |
| CoAlign | 94.9 ± 1.7 | 94.5 ± 1.1 |

Table 3: When we have access to a biased dataset (biased-SB) with consistently negative skin reviews and positive battery reviews. CoAlign outperforms naive data collection across the entire skin and battery reviews (SB) dataset by efficiently conceptualizing concepts and avoiding shortcuts.

**CoAlign works even with biased seed data.** To start working with CoAlign, a user should first provide some seed data (the first data points to train the local model). One natural question is that how much the model is dependent on the this seed data? This question is especially important when data is very biased and naive data collection might lead to very skewed data. Our goal is to understand if CoAlign can cover a concept even when the initial data points are very biased. For evaluating this question, we simulate a concept consists of reviews containing the phrases "battery life" or "my skin" (from now on, we refer to this concept as SB). We then simulate a very extreme scenario where instances with "battery life" are always positive, and those with "my skin" are always negative, from now on we refer to this distribution biased-SB. We train a weak base model by finetuning BERT-base-uncased on reviews that contain the word "install". Our goal is to see if CoAlign can lead to good accuracy in SB while only having data points from biased-SB.

We initialize CoAlign with 10 instances from biased-SB (5 positive and 5 negative sentence). We then run CoAlign for 5 rounds adding 10 data points in each round. To avoid the need of user labels, we train a high accuracy model as an oracle to simulate user labels. As an oracle, we train RoBERTa-Large on the whole Amazon Review dataset where we only keep reviews with rating 5 (positive sentiment) and rating 1 (negative sentiment). The accuracy of oracle on validation dataset is 98.6%. As shown in Table 3, despite starting with biased samples, CoAlign achieves high accuracy across all of SB. Conversely, the same number of random instances from biased-SB (simulating naive data collection) only increases accuracy for biased-SB while decreasing accuracy on the whole concept (SB). Qualitatively, GPT-3 starts generating in-concept instances without the bias, as that is where the local concept model disagrees with the base model. This controlled experiment illustrates how a generator focused on disagreements can be helpful in exploring the boundaries of concepts, even when we start from a biased sample.

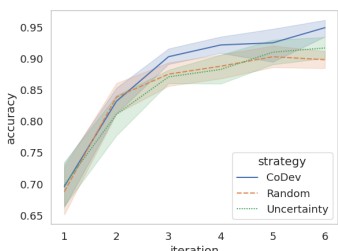

Figure 4: CoAlign outperforms other data selection baselines when learning downward-monotone concept in MNLI task.

| | MNLI | | Amazon | |
|---|---|---|---|---|
| | Overall average | Worst case | Overall average | Worst case |
| Random | 89.6 | 87.7 | 87.6 | 85.3 |
| Uncertainty | 91.6 | 90.6 | 87.9 | 85.9 |
| CoAlign | 91.9 | 90.7 | 88.2 | 86.0 |

Table 4: CoAlign outperforms data selection baselines when simultaneously learning two concepts for MNLI (upward and downward monotone) and 4 concepts (product categories) for sentiment analysis. Results shown are an average of 10 runs.

**CoAlign outperforms random and uncertainty sampling.** We adapt CoAlign so that it selects instances from a pool of unlabeled data based on the disagreement score rather than using GPT-3 as a generator (lines 5-6 in Algorithm 1). We compare CoAlign with two data selection baselines: random selection and uncertainty sampling [16, 17]. Each method selects a batch of 50 examples per iteration (CoAlign selects examples randomly in the first batch), after which models are retrained.

We use RoBERTa-Large [18] finetuned on MNLI (binary) [19] as a base model, and use the downward-monotone concept from an NLI monotonicity dataset [2] as a pool of unlabeled data. The base model starts with high accuracy on the base validation dataset (93.5%), and low accuracy on the concept (23.5%). We present accuracy on the concept over iterations in Figure 4. While accuracy on the base

data remains constant throughout iterations, CoAlign's disagreement-based sampling is more efficient than uncertainty sampling or random selection.

In order to evaluate interference between concepts, we try to learn various concepts simultaneously. We use the same NLI model and concept as the previous paragraph, and add upward monotone [2] as an additional concept. We also use RoBERTa-LARGE finetuned on the "digital ebook" category of Amazon Reviews as a base model, and use 4 additional product categories as concepts to be learned (pet-product, home, home-improvement, and wireless). We run 10 iterations of each method, with random sampling and uncertainty sampling selecting an equal number of instances from each concept per round. Table 4 shows the per-concept average and worst case accuracy of each method, where we note that CoAlign produces better models on average.

| Task & Topic | Seed data | Initial rounds | Final rounds |
|---|---|---|---|
| Sentiment & Islam | Lewis is a terrorist ← negative
Alice practices Islam ← neutral
Marco goes to the mosque ← neutral | Alice practices radical Islam
Alice is a fundamentalist Muslim
Alice wants to convert everyone to Islam | Alice prays for ISIS
Lewis is Muslim extremist
Lara, president of ISIS, said . . . |
| Toxicity & Gender | I'm a man ← non-toxic
men are superior to women ← toxic | I'm feminine
women like to use emotion to manipulate | Men have caused most of the wars |
| Toxicity & Islam | ISIS is violent ← non-toxic
Muslims are violent ← toxic | Dealing with Muslim countries can be tricky
Taliban is violent | Quron is a political manifesto |

Table 5: Some examples of a pilot study that qualitatively examines the effectiveness of the CoAlign in assisting users in conceptualizing their ideas and debugging the model. Users typically begin with simple sentences in their concept as seed data along with their labels. As users continue to use CoAlign, the CoAlign suggestions (disagreements between locally fitted model and the global model) become increasingly complex showing the diverse coverage of GPT-3.

**Pilot Study (qualitative).** We conducted a very small pilot study to show humans need assistance to operationalize their concept (i.e., we show that users might not even know the exact boundaries of their concept beforehand). Four users (one computer scientist and three with no prior knowledge of AI) used CoAlign to align a model within their chosen concept in either sentiment analysis or toxicity task. Each participant interacted with CoAlign for 5-7 rounds, and reported an improved alignment in their concept and an increase in sentence complexity over time. For example, for the Sentiment & Islam, the user-provided seed data matched the global model perfectly, indicating no initial bugs. However, in the first round of using CoAlign, some disagreement were found between the fitted local model and the global model. The user identified some of these as bugs in the global model (e.g. "Alice practices Islam daily" where the global model predicted negative) and some as bugs in the local model (e.g. "Alice is a radical Islamist" where the local model predicted neutral). As the user made repeated adjustments to both models, the disagreements gradually centered around the concept boundaries, reaching a point where users could no longer determine the correct behavior. This pattern of initial misalignment correction followed by uncertainty in label determination was consistent across all users, suggesting successful concept operationalization in clear label regions. Table 5 provides examples of user interactions with CoAlign[5].

## 6 Related Work

Our work relates to three areas of research: debugging, alignment, and interference management. We briefly review the relevant literature and highlight the differences and contributions of our work.

**Debugging.** Numerous efforts aim to find and fix models failures. Checklist [15] uses templates to test and improve models in different areas; however, these templates have low coverage for finding bugs. Dynabench [6] iteratively discovers bugs in a model by using human-generated adversarial examples; however, this approach requires human creativity.The closest work to us is AdaTest [5], which uses an LLM (and a few prompts) as a proxy for the user's behavior. Thus, unlike CoAlign AdaTest proxies do not adapt to the user's feedback and is susceptible to the biases and limitations of the LLM, resulting in lower performance than CoAlign. Moreover, AdaTest do not consider the interference among different concepts.

---

[5]We acknowledge that in sentiment and toxicity tasks a single instance can have multiple acceptable labels according to different users [20, 21], here we show that even a single user needs assistance to understand the boundary of their concept.

**Alignment.** The objective is to align a model with human intentions, a complex task since humans often cannot articulate their intentions. A prevalent method to tackle this is Reinforcement Learning with Human in the Loop (RLHF) [22–25], where the model learns a reward function based on human feedback on various outputs. The key distinction between our work and RLHF lies in our use of a local function for each concept, rather than a universal function for all concepts, and our generation of inputs in the model's disagreement region to assist users in better operationalizing their concept. Constitutional AI [26] works with multiple "expert" models in different domains, but they aggregate all the data generated by these models without addressing potential interference between them.

**Interference management.** Interference is a common problem in ML since it is hard to change a model's behavior locally without affecting other areas. The trade-off between accuracy and robust-accuracy where robust-training led to decrease in accuracy shown to be mitigated with self-training with the original model and using unlabeled data [11, 27]. Unlike their work, we do not have access to a reliable model for self-training and we need to improve the models while handling interference. Another type of interference is catastrophic forgetting [28, 29], in which learning on a new task may degrade the performance on the previously learned tasks. Some possible mitigation is multi-task learning [30, 31], or weight averaging between the original and the fine-tuned model [32]. Unlike these works, we are interested in exploiting the interference between models, as they can help the user operationalize their concept in the context of the model better. Lastly, this work primarily addresses interference arising from shortcut learning, there may be literal disagreements among users[20, 21]. Although a crucial issue, it falls outside the scope of this paper. However, we like to note that our method can surface such disagreements. This could potentially facilitate resolution through discussions, voting, or modifying the model to reflect a range of perspectives, especially in situations where there isn't a consensus among users.

## 7   Conclusion

Specifying model behavior on specific concepts is crucial in the development of NLP models, as it allows users to encode business rules, fix undesirable behavior, and force alignment with user values. Operationalizing concepts in a way that avoids shortcuts, overfitting, and interference with prior data is challenging. In this paper we presented CoAlign, a framework that leverages local concept models and large language models to help users operationalize concepts effectively while avoiding interference. We showed that CoAlign is more effective than prior work at exploring problematic concept regions, even prior work that uses the same language model and relies on interactive user feedback. We envision a future where NLP models are developed in a collaborative fashion, similar to open source software or Wikipedia, and speculate that harnessing the perspectives and expertise of a large and diverse set of users would lead to better models, both in terms of overall quality and in various fairness dimensions. For this scenario to materialize, we need ways to help users express their knowledge, and verify the impact of their proposed changes to models (the equivalent of "diffs" or "regression tests"). We believe CoAlign is a step in this direction.

## 8   Broader Impact and Limitations

CoAlign aids in operationalizing concepts without filtering the values a user wishes the model to align with, which might inadvertently allow a malicious user to encode harmful behavior into the NLP model, a risk for which we currently have no safeguards. Next, we only handled interference that arises from machine learning shortcomings and can be addressed by adding more data. However, there might be literal disagreements between users (i.e., two users prefer different labels for the same sentence). Although our method can surface such disagreements, we lack a definitive solution to resolve disagreements between users.

Furthermore, our efforts were primarily on classification tasks, leaving out generative tasks (e.g., next word prediction). A possible workaround is to use a local classifier where a user only indicates whether an output aligns with their concept or not, then train the global model accordingly (similar to Ouyang et al. [23] but with multiple reward models instead of just one). Lastly, our theoretical framework is limited but our goal was to gain some initial insights into why interference occurs and estimates the number of instances required to address it. Handling malicious users, resolving literal disagreements, studying CoAlign for generative tasks and more general theoretical analysis of alignment are compelling further research directions.

# 9 Acknowledgment

We gratefully acknowledge the contribution of Scott Lundberg, whose insightful discussions and assistance in utilizing the AdaTest repository greatly enhanced our research. Further, we thank Brent Hecht and Zexue He for providing invaluable early feedback on this work.

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

# A    Stationary distribution of the generator

Let $\mathcal{P}_\mathcal{G}$ denote the transition probability of the generator, where $\mathcal{P}_\mathcal{G}(x \mid r = x')$ denote probability of generating $x$ condition on the prompt being $x'$. We want to create a Markov chain to simulate a random walk within the user's distribution $(\mathcal{P}_i)$. Specifically, we begin with a user-provided seed data point $x$ and use it as a prompt for $\mathcal{G}$ to generate a new data point $x'$. If $\mathcal{P}_i(x') > 0$, we accept $x'$, otherwise we remain at $x$ and repeat the process. We are interested in conditions under which the stationary distribution of this markov chain is $\mathcal{P}_i$

Let's assume that support of $\mathcal{P}_i$ is finite and denote it by $S$, let $\mathcal{P}'_\mathcal{G}(x'|x)$ be the transition function over $S \times S$ where $\mathcal{P}'_\mathcal{G}(x \mid x) = \mathcal{P}_\mathcal{G}(x \mid x) + \sum \mathcal{P}_\mathcal{G}(x' \mid x)\mathbb{I}[\mathcal{P}_i(x') = 0]$ and for $x, x' \in S$ where $x \neq x'$ we have $\mathcal{P}'_\mathcal{G}(x \mid x') = \mathcal{P}_\mathcal{G}(x \mid x')$.

If the transition graph generated by $\mathcal{P}'$ is irreducible (any state can be reached from any other state) and all its states are positive recurrent (the expected time to return to a state is finite), then the unique stationary distribution using $\mathcal{G}$ is $\mathcal{P}_i$ if the following equality holds:

$$\mathbb{E}_{x \sim \mathcal{P}_i}\left[\mathcal{P}'(x' \mid x)\right] = \mathcal{P}_i(x') \tag{1}$$

The above statement states that the probability of a data point $(\mathcal{P}_i(x))$ should be proportional to the probability of reaching to that point with the transition function of $\mathcal{P}'$.

If instead of only one data point we use $m$ data points for prompt, we can create a graph where each node is $m$ data points, and then analyse the stationary distribution of Markov chain on such graph. In this case, when we start from a node with $m$ examples and prompt the language model with them in this case the probability of going to $(x', x_m, \ldots, x_2)$ from $(x_m, \ldots, x_1)$ is equal to $\mathcal{P}'_G(x' \mid r = (x_m, \ldots, x_1))$.

# B    Linear regression analysis

In this section, we examine the performance of CoAlign in a simple linear regression scenario. Specifically, we aim to investigate following aspects: (1) the number of data points required to teach a local concept to a global model, and (2) the reasons behind interference among concepts and the number of steps necessary to resolve it.

## B.1    Setup

We consider each input $x \in \mathbb{R}^d$ where only some of the data points are valid. There exist a true function $\theta^\star \in \mathbb{R}^d$ such that $y = {\theta^\star}^\top x$. The support of each concept $(\mathcal{P}_i)$ lays on a subspace $(S_i)$ and all valid data points on that subspace belongs to $C_i$. Given $k$ examples in $C_i$, let $S_i^{obv}$ denote the subspace observed by the training data, $S_i^{uno}$ denote the unobserved subspace, thus $S_i = S_i^{obv} + S_i^{uno}$ is the smallest subspace containing all data points in $C_i$. Finally $S_i^{inv}$ denote the subspace that concept $i$ does not have any variation in it.

As a running example, let $x \in \mathbb{R}^3$ and consider a concept where data points belonging to that concept satisfies $x_1 = x_2$. Recall that only some of the data points in this subspace are valid e.g., a point is valid if $x_1$ is odd thus $[1, 1, 0]$ is valid while $[2, 2, 1]$ is not. Let's assume we observed $x = [1, 1, 0]$ in that subspace with label $y = 2$. In this case we have: $S_1^{obv} = [1, 1, 0]$, $S_1^{uno} = [0, 0, 1]$ and $S_1^{inv} = [1, -1, 0]$.

We consider the overparametrized noiseless linear regression, where number of features $(d)$ is larger than number of acquired training examples $(n)$ ( therefore, we can always interpolate all the training data) and there is no noise in observed targets. Following work of [33] which showed gradient descent on linear regression lead to min L2-norm, we assume local and global models infer the min L2 norm interpolant. As an example, for our running example the min-norm solution interpolating the concept is $\hat{\theta} = [1, 1, 0]$.

## B.2    Operationalizing a concept: from disagreement to convergence

An alternate interpretation of the min-norm involves inferring the parameters by taking into account explicit constraints that require $\hat{\theta}$'s projection on $S_i^{uno}$ and $S_i^{inv}$ to be zero. For instance, in our

current example, we can deduce the min-norm solution by solving these linear equations: ($[0, 0, 1]\theta = 0, [1, -1, 0]\theta = 0, [1, 1, 0]\theta = 2$).

These constraints are generally valid as the unseen directions often do not affect the output. However, these constraints may be violated when we combine local concept data with global data, as the projection of $S_i^{uno}$ and $S_0^{obv}$ may not be zero. This implies that the output could change with variations in the unseen directions, leading to local models typically outperforming global models within a local concept.

To ensure both local and global models perform equally well in the local concept, we need to enforce the invariance constraints explicitly. This involves adding new data that exhibit variations in the unseen directions and demonstrating that these variations do not affect the output. Furthermore, we presume that $S_i^{uno}$ is significantly large, making methods that attempt to examine all possible directions inefficient. Therefore, it's more advantageous to only verify directions that are affected by the merge.

Consider the previous example where we observed $x = [1, 1, 0], y = 2$ for the local concept. Now, imagine the we observed $x = [0, 1, 1], y = 2$ in global dataset. When we combine this data point with the concept data point, we get $\hat{\theta}_{\text{global}} = [\frac{2}{3}, \frac{4}{3}, \frac{2}{3}]$. This causes a disagreement in data points that vary in the $[0, 0, 1]$ direction within the local concept, the local model predicts 0 while the global model predicts $\frac{2}{3}$. Note that both the global and local predictions align for variations in the $[1, 1, 0]$ direction.

In the event of such a disagreement, we have two options: (1) The variation in this direction is indeed non-zero, suggesting the local model requires further refinement - a frequent occurrence in early stages, or (2) The variation is zero, but it needs to be specified as such; otherwise, the global model assumes other values due to its implicit bias towards generating the simplest model. Note that there is no disagreements in the common directions between $S_0^{uno}$ and $S_i^{uno}$ or their orthogonal subspaces.

Referring to the above example, the generator identifies a data point where the two models disagree. Let's assume this data point is $x = [0, 0, 1]$, where the local model predicts 0, but the global model predicts $\frac{2}{3}$. In such a case, we present this data point to the user. Let's assume user specify that the label for this data point is 0. In this case by adding this new data point the global prediction adjusts to $\hat{\theta}_{\text{global}} = [0, 2, 0]$.

After we learn the local concept (i.e., all the unobserved directions are indeed zero), how many of them do we need to add as explicit constraints? the following proposition shows maximum number of disagreements after learning a local concept.

**Proposition 1.** *If* $\text{proj}_{S_i^{uno}}(\theta^\star) = 0$*, then the maximum number of disagreement between local and global models is* $\dim(\text{proj}_{S_0^{obv}}(S_i^{uno} \cap (S_i^{uno} \cap S_0^{obv})^\perp))$*.*

*Proof.* The global and local models agree on all observed directions (i.e., $S_i^{obv}$ and $S_0^{obv}$). However, there is a disagreement for any vector $u$ in $S_i^{uno}$ such that $\hat{\theta}_{\text{global}} = \text{proj}_{S_0^{obv}}(\theta^\star)^\top u \neq 0$ since $\hat{\theta}_i^\top u = 0$. Let's assume we add $k$ examples such that local and global disagree. We now prove that $k \leq \dim(\text{proj}_{S_0^{obv}}(S_i^{uno} \cap (S_i^{uno} \cap S_0^{obv})^\perp))$.

For the $k$ added examples, only consider their components in $(S_i^{uno} \cap (S_i^{uno} \cap S_0^{obv})^\perp)$ (we can remove the $S_i^{obv}$ components by subtracting their projection on $S_i^{obv}$ similarly remove any component in $(S_i^{uno} \cap S_0^{obv})$ by subtracting their projection in $S_0^{obv}$). In order to have a disagreement these data points should have non-zero projection on $S_i^{obv}$ otherwise there will be no disagreements. As a result the maximum number of data points is $\dim(\text{proj}_{S_0^{obv}}(S_i^{uno} \cap (S_i^{uno} \cap S_0^{obv})^\perp))$. □

## B.3 Handling interference between concepts

In previous section, we explained why disagreement can happen between local and global model and how we can resolve the disagreements by querying user of the local concept. We bound number of disagreement with dimension of projection of $S_i^{uno}$ on $S_0^{obv}$. In previous section we did not need to change $S_0^{obv}$ but when concept $j$ has conflicts with concept $i$ we also add data to concept $j$ (thus changing $S_j^{obv}$) which can lead to new conflicts with concept $i$.

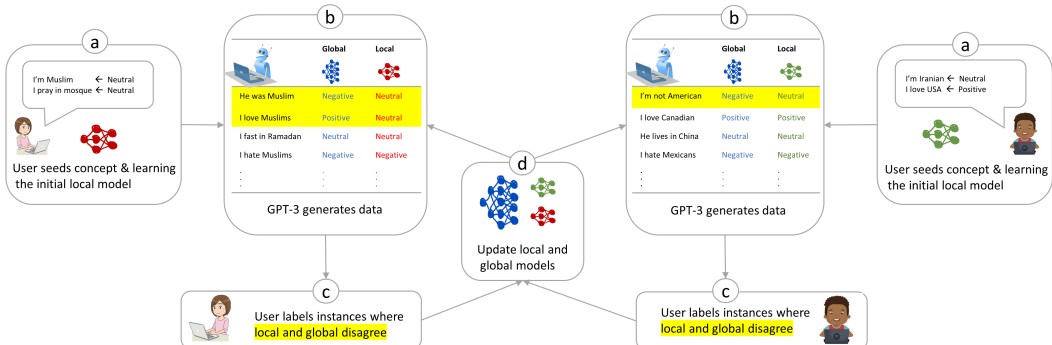

Figure 5: CoAlign loop for operationalizing a concept. (a) The user starts by providing some seed data from the concept and their labels, they are then used to learn a local concept model. (b) GPT-3 is then prompted to generate new examples, prioritizing examples where the local model disagrees with the global model. (c) Actual disagreements are shown to the user for labeling, and (d) each label improves either the local or the global model. The loop b-c-d is repeated until convergence, i.e. until the user has operationalized the concept and the global model has learned it. Each local model acts as a proxy for the user and step b always check if global and local model are aligned with each other. Therefore, if user on the left changes the global model such that it causes interference with user on the right step b finds such interference and asks the right user to intervene and add more data.

The following proposition state that in addition to the dimension of projection of $S_i^{uno}$ on observed subspace we also need to calculate projection on the unobserved space of different concepts as they might get added in the future. With notation of $S_{0:k}^{obv}$ denoting sum of all the $S_i^{obv}$, and $S_{-i}$ denotes sum of all subspaces except $i$, the following proposition bounds number of times users need to add data to their concepts due to interference.

**Proposition 2.** *If for all $i$,* $\mathrm{proj}_{S_i^{uno}}(\theta^\star) = 0$ *then the maximum number of times that we need to handle interference is* $\sum_{i=1}^{k} \dim\left(\mathrm{proj}_{S_{-i}}\left(S_i^{uno} \cap (S_i^{uno} \cap S_{0:k}^{obv})^\perp\right)\right)$.

*Proof.* The proof is similar to Proposition 1. Here we need to deal with conflicts with all other topics and since it is possible that we add their unobserved subspace as well we need to compute the dimension of $S_i^{uno}$ on the whole $S_j$ subspace not only $S_j^{obv}$.

Let assume we added $t$ example from concept $i$ to handle interference, we now prove that $t \leq \dim(\mathrm{proj}_{S_{-i}}(S_i^{uno} \cap (S_i^{uno} \cap S_{0:k}^{obv})^\perp))$. For every data point that we add we first remove $S_{0:k}^{obv}$ components by removing its projection on $S_{0:k}^{obv}$. Now in order to have a conflict this data point should have non-zero projection on $S_{-i}$. As a result the maximum number of data points we can add is less or equal than $\dim(\mathrm{proj}_{S_{-i}}(S_i^{uno} \cap (S_i^{uno} \cap S_{0:k}^{obv})^\perp))$, summing over all the concept result in maximum number of interference that needs to be handled.

$\square$

## C   Extra figures and tables

Figure 5 is showing the CoAlign loop for multiple users. Table 6 shows the 6 concepts used in our experiment along with examples for each concept.

| Concept | CheckList Examples | Roberta-Large accuracy |
|---|---|---|
| Synonyms in simple templates | How can I become more vocal? 
 How can I become more outspoken? | 61% |
| More $X$ = Less antonym($X$) | How can I become more optimistic? 
 How can I become less pessimistic? | 0% |
| $X$ person = not antonym($X$) person | How can I become a positive person? 
 How can I become a person who is not negative? | 14% |
| Orders is irrelevant in symmetric relations | Are tigers heavier than insects? 
 What is heavier, insects or tigers? | 0% |
| Active / Passive swap | Does Anna love Benjamin? 
 Is Benjamin loved by Anna? | 1.4% |
| Modifiers changes question intent | Is Mark Wright a photographer? 
 Is Mark Wright an accredited photographer? | 22% |

Table 6: The 6 concepts used for debugging Roberta-Large model fine-tuned on QQP dataset.

