# OpenReview forum: "Collaborative Alignment of NLP Models"
_NeurIPS.cc/2023/Conference — NeurIPS 2023 poster_

### Official Review · Reviewer_g41d · 2023-07-04

**Soundness:** 2 fair
**Presentation:** 2 fair
**Contribution:** 2 fair
**Rating:** 5
**Confidence:** 5

**Summary:**

This paper frames operationalizing concept as a solution to enumerate all possible concepts and developed a corresponding framework CoDev, that starts by collecting text and labels from users, then have GPT-3 generates text and labels. When there are disagreement among users and GPT-3, users would be asked to relabel the instance until convergence. The authors conducted experiments using Amazon review dataset and MNLI to compare CoDev with baseline models. Also the author conduct a pilot study with 4 people using CoDev from providing data to improve model alignment and more complex output after 5-7 iterations.

**Strengths:**

The paper points out an important questions that NLP model align with multi-users values. And it is novel to use GPT-3 as the global model and Roberta-large trained on users input as the local model, and try to converge the disagreement from two models through collecting user labels iteratively.

**Weaknesses:**

1. Given the paper, that the authors handle disagreement by repeating collecting user labels until convergence, assumes that there exist a single ground truth for each instance. But many studies have shown that a single instance can have multiple acceptable answers [1, 2]. The study design also conflict with the study goal of aligning multi-users' value. Disagreement is not a sign of error but could be a sign of multiple possibility. How could you determine the disagreement between the local model and the global model is not acceptable?

[1]Wan, R., Kim, J., & Kang, D. (2023). Everyone’s Voice Matters: Quantifying Annotation Disagreement Using Demographic Information. Proceedings of the AAAI Conference on Artificial Intelligence, 37(12), 14523-14530. https://doi.org/10.1609/aaai.v37i12.26698
[2]Aida Mostafazadeh Davani, Mark Díaz, Vinodkumar Prabhakaran; Dealing with Disagreements: Looking Beyond the Majority Vote in Subjective Annotations. Transactions of the Association for Computational Linguistics 2022; 10 92–110. doi: https://doi.org/10.1162/tacl_a_00449

2. The authors mentioned the shortcoming of having expert label datasets in the introduction. But actually many open sourced dataset for machine learning studies are labeled by crowd workers from third party platforms such as Amazon MTurk. And the determination for sentiment or toxicity is a very subjective task, that is there is no expertise to judge whether others feels the text is toxic or not. Therefore, the research gap mentioned in the introduction and the dataset/task picked for the experiment make author's framework and argument weak.

3. For table 2, it is not clear how SB is different from biased SB. And it is also not clear how Data collection method is different from base line BERT model and CoDev model.  And many clarity concerns as listed in the following Question section. These confusions make the paper less convincing.

**Questions:**

1. There is no clear definition for your local model and global model other than the figure 1. Could the authors please clarify why and how they use GPT 3 as a global model? What prompts are used?

2. It is not clear whether the d step (repeating labeling) in the Figure 1 is involved with a single user during the iteration or multiple users. If multiple users, how did the authors handle the disagreement among users? If that's different user for different round, this need to be clarified. If single user,  how could the CoDev system enable multi-user interaction?

3. For the pilot study, it is not clear how the authors collect the seed data. Are those labels given by single user for each instance? If each instance collect labels from multiple users, how does the authors aggregate the labels?

4. For the pilot study, it is is not clear how the authors evaluate their study. If better alignment is the evaluation metric, how do the authors evaluate the quality of alignment? In the line 289 and 290, the authors said 'As the user made repeated adjustments to both models, the disagreements gradually centered around the concept boundaries, reaching a point where users could no longer determine the correct behavior.' It is not clear why users could no longer determine the correct behavior is the standard for good alignment. Shouldn't users always have the right and ability to determine what they think is correct?

5. In Table 4, why is there only labels for the 'Seed data' column but the 'Initial rounds' and 'Final rounds' don't have corresponding labels? Are the Initial rounds and final rounds evaluate alignment based on the labels? And the Table 4 caption mainly highlight the text in final rounds become much complex than the initial rounds. Why is this important? Is this an evaluation criteria for the pilot study?

**Limitations:**

The authors discussed the limitation of solving literal disagreement since they only handled interference caused my machine learning shortcoming.

---

> ### Author Rebuttal · Authors · 2023-08-10
>
> Thanks a lot for your comments and constructive feedback.
>
> W1) As mentioned in the general rebuttal we agree that literal disagreements and multiple possibilities is a very important question but even in the absence of literal disagreements, the problem of ML interference still remains and that is the main focus of this paper. We appreciate your comment and references and we will add an extra discussion section to address this issue.
>
> W2) We agree that many open sourced dataset are labeled by Amazon turk however, we are focusing on post-training adjustment. For example in the NLI task, the dataset is collected through third party labeling but further studies showed that it has low performance in concepts such as downward and upward monotonicity. In response, some papers collected data to address the low performance in these concepts but it can easily be shown that the whole concept is not covered.
>
> Furthermore, collecting data and labeling is very challenging for niche concepts that collecting data is hard. Our work uses LLMs to address such issues. In other words, we use LLMs as an infinite pool of unlabeled data and CoDev helps users to find and label data points in this pool that belong to their concept.
>
> We totally agree with you that tasks such as sentiment and toxicity are very subjective tasks, and different users might have different opinions; the goal of pilot study on such tasks was to show that even a single user needs assistance to operationalize their concept (more details are below in q3-5).
>
> W3) We apologize for the ambiguity of text for this section, we will improve the writing in camera-ready. In summary, as stated CoDev uses LLM to generate data for a concept. To start the process users need to provide some seed data, the point of this experiment is to show that even in the very extreme case of biased seed data, CoDev can still generalize and cover the concept.
> - SB here is all reviews containing the word “skin” or “battery”
> - biased-SB contains reviews that are positive and have word “skin” in them; or that are negative and have word battery in them.
> - Base model is a model that has low performance in reviews containing skin and battery and our goal is to improve the performance of this model
> - Normal data collection by definition only adds data that are biased (i.e., reviews that are positive and have word “skin” in them; or that are negative and have word battery in them). As a result, data collection leads to very high performance in biased-SB but low performance in SB.
> - CoDev on the other hand improves performance in both SB and biased-SB; even though it started from biased seed data.
>
> ---
>
> Q1) Both local and global models are classifiers (roberta-large in our experiments). GPT-3 is used for generating sentences within a concept. Lines 94-96 explain the process. In particular, our goal is to find sentences that belong to a concept. In order to do so we iteratively use m examples from the concept as a prompt to GPT-3. In response GPT-3 generates more examples within the concept. We apologize for the confusion and will make the definition more clear.
>
> Q2) For simplicity, Figure 1 is only showing the steps in CoDev for a single user. In section 3.4 we explain interference and have we handle it. We will add a new figure with multiple users in the appendix for clarity.
>
> Q3) For each row, the seed data is written by a single user in his/her own concept.
>
>
> Q 4-5) This pilot study is a small part of our experiment section, our goal for conducting this small pilot study was to show:
> - Humans need assistance to make their concept clear. For example in toxicity - islam case, the user thought they have a good grasp of the concept in their mind. However when the user states that “ISIS is violent” is non-toxic but “muslims are violent” are toxic; CoDev asked the user about Taliban, Hezbollah, radical Islam, etc. and as shown in Table 4 this back and force resulted in sentences that user has not think about such edge cases beforehand and could not come up with a label (i.e., require more thinking to make their concept clear).
> - As users continue working with CoDev, the number of disagreements between local model and global model decreases and gets limited to sentences that users might not count as clear misalignment to their concept.
> For more thorough analysis of CoDev please refer to other parts of our experiment section.
>
> We believe that helping users to operationalize their concepts and handling ML interference are important challenges in alignment and we hope these comments address the listed weaknesses enough and you consider changing the score.

---

> > ### Comment · Reviewer_g41d · 2023-08-17
> >
> > Thank you for the thorough response and clarification. Please consider integrating these clarification, explanations and the discussion into the manuscript. After reading the rebuttal, I decided to change the score from 3 to 5.

---

### Official Review · Reviewer_rcBm · 2023-07-06

**Soundness:** 3 good
**Presentation:** 3 good
**Contribution:** 3 good
**Rating:** 6
**Confidence:** 3

**Summary:**

This paper proposes a new framework to debug NLP models. Specifically, while debugging a global model, it starts with training some local models on specific concepts. Then new data used for model improvement is labeled if the global model disagrees with the local model.

Experiments indicate that the proposed approach (CoDev) outperforms the baseline approach (AdaTest) over multiple settings.

**Strengths:**

1. This work focuses on an important but under-explored problem: debugging NLP models. The findings could potentially benefit the community.
2. The paper is well-written, easy to understand the contents.

**Weaknesses:**

1. There is a gap between theoretical guarantee and real experiments. Specifically the paper experiments with LLM, but provides proof based on linear regression.
2. It's unclear if some of the results are statistically significant. For example, in Table 3 (about MNLI and sentiment analysis), the improvement seems marginal.
3. All experiments seem to focus mainly on toy tasks like classification or NLI. The takeaway would be stronger if the paper has discussions about extending to more challenging NLP tasks such as structure prediction.

**Questions:**

1. Missing discussion about active learning in related work.
2. (see weakness 3) How to extend the proposed framework to structure prediction tasks? Also, which NLP tasks do you think the proposed approach would fail?


**Limitations:**

Yes, this paper includes the limitation section.

---

> ### Author Rebuttal · Authors · 2023-08-10
>
> Thank you so much for your constructive feedback.
>
> - W1) Unfortunately providing theoretical results for deep neural nets (and transformers in our case) is an open problem however, previous work (reference 10,11,12,13) consider overparameterized linear regression as a way to provide some insights on how these models might work. We followed this line of work and showed how our method works in overparameterized linear regression settings. The whole point of the theoretical section is to show that similar to our intuition, if learning the desired function in a local neighborhood requires a few samples then we can learn a concept in isolation with few queries (instead of asking users a lot of queries, e.g., look at fig2(c)). However adding other concepts can cause interference (e.g., see fig2(b)). Our theory bounds how many new queries we need to ask the user due to interference (as shown in proposition 1,2, number of queries can be a lot smaller than the dimension of inputs (d)).
>
> - W2) We ran each experiment in Table 3, 10 times and reported the average. We ran the t-test on the distributions although the improvement is small but the t-test shows statistical significance. We will provide the t-test results in camera-ready.
>
> - W3) For non-classification tasks such as next word predictions we can assume that the local functions are a mapping from pair of (input, output) to a binary label of if the output is desired or not (e.g., (all muslims are, murderer -> non-desired) but (all muslims are, going) -> desired). Therefore, the local functions can still be binary and simple while the global function can be complex (e..g, generation instead of classification). Thank you so much for this comment and we will add a discussion on how to extend this work to more sophisticated non-classification tasks. We also like to note that we have experiments on multi-class classifications in our experiments, not just binary.
>
> ---
>
> - Q1) We appreciate your comment and will add an extra section on comparing with active learning methods. In summary, CoDev learns each concept individually in isolation and only query inputs where global and local disagree. Thus it does not need to query part of space that local and global overlap (i.e., agree). Also learning local concepts in isolation allows CoDev to better learn the local concept in comparison to active learning methods that only adds data to the global model and might only memorize the local concept data.
>
> - Q2) We explained how to extend this work to more complex scenarios in (W3). This approach fails for a user that considers a very general concept that cannot be divided into subparts.
>
> We hope these comments address the listed weaknesses enough to warrant a change in score.

---

### Official Review · Reviewer_q4QC · 2023-07-07

**Soundness:** 3 good
**Presentation:** 3 good
**Contribution:** 3 good
**Rating:** 6
**Confidence:** 1

**Summary:**

This paper describes a multi-user collaborative model alignment framework that teaches certain desired concepts (behaviors, rules) to large language model (LLM).
The authors train a global model that intergrates the original data and all concepts, and a local model for each concept. The LLM is guided to generate examples where the local and the gloabl model disagree, and these examples are presented to users for annotation. Finally, these new annotations are used to update local and global models.

The proposed framework is evaluated on cases that teach an NLI model about downward- and upward-monotone concepts and a sentiment classifier about Amazon products reviews.
Results show that it outperforms AdaTest and other baselines.


**Strengths:**

* the targeting question is an important question and has attracted a lot of research interest. the proposed approach is easy to understand and experimental results show that it is effective, especially when there is interference between multiple concepts

**Weaknesses:**

* the experimental setup is not described in detail, it may not be easy for other researchers to reproduce the study


**Questions:**

Line 29: 'does not to lead to'

---

> ### Author Rebuttal · Authors · 2023-08-10
>
> Thanks for your constructive feedback. As mentioned in the general rebuttal, we will release code and data in the camera ready, and we will make the experiment section more clear. Hopefully this information removes this weakness, leading to an improved score.

---

> > ### Comment · Reviewer_q4QC · 2023-08-13
> >
> > I have read the rebuttal and other reviews.

---

### Official Review · Reviewer_MmNy · 2023-07-10

**Soundness:** 2 fair
**Presentation:** 2 fair
**Contribution:** 3 good
**Rating:** 6
**Confidence:** 4

**Summary:**

The paper proposes a method based on data augmentation and instance selection for training a supervised model to be aligned with "concepts", where concepts dictate specific model behavior on certain inputs. In the proposed setup users illustrate a concept with a training example, this is followed by generating additional examples with GPT3, the acceptance of the generated examples is proportional to the disagreement between a global model trained on all the data and a local model trained to learn the concept perfectly. In experimental evaluations the proposed instance selection method seems to outperform reasonable other strategies for selecting data for aligning supervised models on specific concepts.

**Strengths:**

- The proposed data selection method seems original, the presented method with an LLM data generator in the loop is under explored.
- The problem of enforcing user desires on a supervised model is meaningful.


**Weaknesses:**

- The proposed method for aligning user "concepts" seems somewhat laborious or frustrating from a user-stand point, especially in cases where users concepts conflict with those of others. E.g. the paper notes, "In practice, this means that a user adding a new concept needs to make sure it does not break any concepts from other users, a process similar to regression testing in software engineering." -- What happens in cases where the users concept cannot be incorporated without breaking somebody else's concept?
- The experimental setup of the paper is hard to follow and is unlikely to be reproducible.

**Questions:**

- The idea of a concept seems a little vague; please consider discussing what kinds of user intents may be captured in the concepts. Perhaps with an eye toward realistic concepts that users want to see alignment on. As I see it, the concept seems like a rule for labeling specific input instances based on easily examinable features of the input.
- Many of the ideas described in the paper (user-defined concepts, conflicts between then, training on user concepts) seem similar to the line of work on programmatic creation of datasets with weak labelers. Please cite and discuss similarities and differences to this line of work: https://scholar.google.com/citations?user=rfwwtFYAAAAJ&hl=en
- The paper (through references of similarity) and especially the experimental setup (first two sub-parts) heavily depend on familiarity with prior work. Please describe the experimental setup and these prior works in greater detail in this paper.
- This is perhaps a matter of style and framing, but I would recommend renaming the title of this paper. The presented work seems to be much more specific in its scope than the title seems to indicate. The presented approach relies on users labeling data (significantly devalued labor in ML) to align models with their preferred concepts, a large body of work may be framed as "collaboratively developing NLP" models by this naming logic. Further, the explored tasks are simplistic classification tasks; arguably, NLP encompasses a significantly broader range of tasks than these alone.

**Limitations:**

The paper discusses limitations adequately.

---

> ### Author Rebuttal · Authors · 2023-08-10
>
> Thanks a lot for your constructive feedback.
>
> W1) In this work we focused on post-training adjustment to enforce business rules, rectify undesired behavior, or align with user values.  A concept relates a set of inputs to desired behaviors, e.g. “religion does not connote sentiment”. For teaching a model about a concept we should create inputs as well as outputs. As you mentioned one way to do so is to write a program to automatically generate a dataset for a concept. However, concepts can be more complex and writing some programs cannot cover the whole input space. For example, Checklist (reference 14) is similar to your references of programmatic creation of data. In particular, Checklist creates a dataset with some rules for each concept. However as shown in the first part of experiment section (Table 5) these programmatic patches will miss a lot of subtleties for these concepts (i.e., after teaching a concept to a model we can simply run CoDev to find a lot of bugs that have not been covered). CoDev on the other hand utilizes GPT-3 to generate inputs that cover the concept more thoroughly. In other words, writing a program to generate inputs only covers a small part of the concept while random walk using GPT-3 as described in 3.1 can cover the majority of the concept space.
>
> Furthermore, as shown in the pilot study, labeling of inputs from a concept can be tricky and cannot be written with a simple program. For example, in the toxicity-islam case in table 4 the user starts with stating that “ISIS is violent is non-toxic” but “Muslims are violent is toxic” and then CoDev asks the user about sentences like “Taliban is violent”, or “Hezbollah is violent”, etc. Thus the user should interact with the model to make the labeling more clear, this cannot be done with simple programs.
>
> Regarding Interference, a user wants to add their concept without breaking the model on other concepts. The presence of possible conflicts is annoying, but that is not a function of our method, it's a function of the real world complexity and our method helps solve it. The alternative of not having explicit concepts and just labeling data would not work and would be more frustrating than handling interferences. For literal disagreement between users please see the general rebuttal.
>
> We also like to note that our method making the concept alignment less laborious for the user by generating sentences using GPT-3 and only requiring user to label them, also the majority of alignment checks is happening between local and global model and user is only queried when there is a disagreement between local and global function (section 3.3).
>
>
> W2) As mentioned in the general rebuttal we are going to release code and data for reproducibility. We will also make a significant improvement on the writing of the section.
>
> Q1) We addressed this question in (w1). Also as shown in the experiments concepts can be things like downward or upward monotonicity in NLI (a well-known problem in state of the art NLI models), or simpler concepts such as “X person = not antonym(X) person“ in QQP that are revealed by [14] that SOTA models performs really bad on them (table 5 in appendix) or even more complex concepts as depicted in the pilot study.
>
> Q2) Checklist (reference 14) is similar to programmatic creation of data. In particular, Checklist creates a dataset with some rules for each concept. However as shown in the first part of experiment section (Table 5) these programmatic patches will miss a lot of subtleties for these concepts. CoDev utilize GPT-3 to align the model with the concept completely and also handle interference between concepts. Thanks a lot for the references, we will make a more thorough comparison to programming creating datasets with weak labelers in camera-ready.
>
> Q3) Thanks a lot for the feedback, we will make the first two sections of the experiment setup more clear for the camera-ready.
>
> Q4) Thanks for the feedback we will make the title more specific.
>
> We hope these comments ameliorate the weakness listed enough to warrant a change in score.

---

> ### Comment · Reviewer_MmNy · 2023-08-20
> **Acknowledgement of rebuttal.**
>
> I have read the author rebuttals and other reviews. The rebuttals are illustrative, and I encourage the authors to make the changes they have commited to in their response.
>
> I stand by my original assessment of the paper: It presents a well-scoped novel method and is well-executed, however, it presents work that is no more than moderate-to-high impact in my best judgment.

---

### Official Review · Reviewer_11Xh · 2023-07-12

**Soundness:** 2 fair
**Presentation:** 3 good
**Contribution:** 3 good
**Rating:** 6
**Confidence:** 1

**Summary:**

The authors propose a framework for collaborative NLP development (CoDev) that enables multiple users to align a model with their beliefs.
The proposed work CoDev aids users in clarifying their concepts (an area humans often struggle with) and assists ML models to handle conflicts between concepts (an area ML often struggle with due to its inability to accommodate local updates).

The authors' main insight is learning a local model for each concept, and a global model to integrate the original data with all concepts. The authors then steer a large language model to generate instances within concept boundaries where local and global disagree. Their experiments show CoDev is effective at helping multiple users operationalize concepts and avoid interference for a variety of scenarios, tasks, and models.



**Strengths:**

1. The paper is pretty easy to follow and propose a new framework that is useful for controlling LLMs to constrain the outputs according to user defined concepts.
2. The idea of creating proxies from local models for user concepts is novel and interesting.
3. The experiments with real world users is nice, as this work is all about collaborative development from users.
4. There are some theoretical analysis in the proposed framework which is nice and provides additional insight.

**Weaknesses:**

1. I think more discussion and datasets could be used in terms of CoDev with biased seed data and CoDev with unlabeled data. Since the work is very central about the data distribution and checking for controlling models via user input, I think a more diverse type of dataset could be used, instead of just selecting positives and negatives from review dataset. It might provide better evidence in generalizability of such framework.
2. Since AdaTest is a major related work, maybe some more comparison with it in the seeded datasets and unlabeled data would be nice?

**Questions:**

Above.

**Limitations:**

The limitations and societal impact are properly discussed.

---

> ### Author Rebuttal · Authors · 2023-08-10
>
> Thank you so much for your constructive feedback.
>
>
> - W1) We chose only positive and only negative cases to showcase that even in a very extreme case of bias in the seed data, CoDev can still generalize to the whole concept. We will add extra experiments with different distributions such as other segments of this dataset with less extreme biases.
>
>
> - W2) Adatest iteratively uses 7 examples from a concept to generate more examples, and thus it cannot work with unlabeled data.  As a result we cannot compare CoDev against Adatest when unlabeled data exists. We will make the difference between AdaTest and CoDev more clear in the camera-ready. We note that where we can, we do compare to AdaTest (first section of the experiments), both for finding bugs and fixing bugs without interference (Table 1 and line 225-229).
>
>
> We hope these comments ameliorate the weakness listed enough to warrant a change in score.

---

> > ### Comment · Reviewer_11Xh · 2023-08-18
> >
> > I have read the rebuttal and other reviews.
> > Thank you very much for the explanation as they help clear things up a bit.

---

### Author Rebuttal · Authors · 2023-08-10

We like to thank the reviewers for their constructive feedback, and stating that the problem we are considering is “important” and “beneficial to the community”, also stating that our “approach is novel”, our “theory provides some insights”, the paper is “well written” and “experiments show the effectiveness the method”. Following we are mentioning some of the concerns shared by the reviewers.For other concerns please see individual responses.

---
## Literal interference between users:
---
Reviewers 2 and 5 have pointed out that our work does not address literal interference, where two users assign different labels to the same sentence. We acknowledged this limitation in the paper and highlighted it as an avenue for future research. We will make this more clear in the revision, and note that the interference we focused on still remains as an important problem even if there is no disagreement between users.

To clarify, there are two distinct types of interference when aligning a model to multiple users (or training a model on different distributions):
1. Literal interference: where users disagree in labeling the same sentence.
2. ML interference: interference arising from over-generalization in machine learning.


Both challenges are critical, yet they address different facets of the problem. Our paper particularly addresses the latter, ML interference. Previous research [1,2,3] has shown that any local change in ML models can interfere with other parts of the model (e.g., comparing fig3(a) and fig3(e) you can see that a change inside the concept caused a change outside the concept). We want our model to not overfit (i.e., only memorize the training data) and generalize to unseen data; however this generalization is very dependent on other concepts and the previous data. Take, for instance, a world where only bananas are yellow. An ML model might be guided by a user to recognize bananas solely based on their yellow color. Now if a new user introduces another yellow object, like corn, the model must discern other distinguishing features. Merely combining training data for two such concepts doesn't suffice (as shown in [2]); the boundaries must be distinguished. Our work outlines methods to mitigate this type of interference.


Finally we like to note that while we haven't introduced a specific mechanism to resolve literal interference, our method can surface such interference. This can pave the way for resolution through discussions, voting, or even tweaking the model to reflect multiple perspectives, especially in cases where, as R5 noted, there isn't a consensus among users.

We apologize for the ambiguities in text. In the new version we delineate both types of interference, emphasizing the intricacies of ML interference. We will also add an experiment to surface disagreements between two local models (i.e., proxies for humans) to showcase the effectiveness of our method to surface disagreements. We again acknowledge that handling literal disagreement is very important but out of scope of this work and great future direction.

---

## Some ambiguity in the experiment section and reproducibility of results
---
Reviewers have mentioned that some parts of the experiments are hard to follow and might not be reproducible. We are releasing the full code and data in camera ready, so the experiments can be easily reproduced. Even though some experiments have humans in the loop and thus inherent variance, the gap between AdaTest and Codev (Table 2 and line 226) is large enough that this should not matter.

Regarding the ambiguity we will make a huge improvement on writing and apply all your feedback in text. In summary the following are the message of the four main section of our experiments:

- CoDev works better than AdaTest by finding more bugs and causing no interference (in comparison to AdaTest that causes interference).
- CoDev works even when the seed data is biased
- CoDev sampling mechanism outperforms random or uncertainty sampling
- A very small pilot study to show humans need assistance to operationalize their concept (i.e., we showed that they might not even know the exact boundaries of their concept beforehand).


---
[1] Khani, Fereshte, and Percy Liang. "Removing spurious features can hurt accuracy and affect groups disproportionately." Proceedings of the 2021 ACM conference on fairness, accountability, and transparency. 2021.

[2] Raghunathan, Aditi, et al. "Adversarial training can hurt generalization." arXiv preprint arXiv:1906.06032 (2019).

[3] Srivastava, Megha, et al. "An empirical analysis of backward compatibility in machine learning systems." Proceedings of the 26th ACM SIGKDD International Conference on Knowledge Discovery & Data Mining. 2020.

---

### Decision · Program_Chairs · 2023-09-21

**Decision:**

Accept (poster)

**Comment:**

This paper presents a framework for multiple users to interact with a large language model, such that users can operationalize "concepts" and try to get the LLM to align to their concepts. This is done with the structure of having a global model of all concepts, as well as a local model for each of the concepts.

This problem is interesting, and the approach seems sound and well-carried out. There were several promises of substantial revision during the author-reviewer discussion. I strongly urge the authors to follow through on these revisions.